# Intrapatient Comparison of Coblation versus Electrocautery Tonsillectomy in Children: A Randomized, Controlled Trial

**DOI:** 10.3390/jcm11154561

**Published:** 2022-08-04

**Authors:** Kyu Young Choi, Jae-Cheul Ahn, Chae-Seo Rhee, Doo Hee Han

**Affiliations:** 1Department of Otorhinolaryngology-Head and Neck Surgery, Kangnam Sacred Heart Hospital, Hallym University College of Medicine, Seoul 07441, Korea; 2Department of Otorhinolaryngology-Head and Neck Surgery, CHA Bundang Medical Center, CHA University, Seongnam 13496, Korea; 3Department of Otorhinolaryngology, Seoul National University Hospital, Seoul National University College of Medicine, Seoul 03080, Korea; 4Graduate School of Immunology, Seoul National University College of Medicine, Seoul 03080, Korea; 5Institute of Allergy and Clinical Immunology, Seoul National University Biomedical Research Center, Seoul 03080, Korea; 6Sensory Organ Research Institute, Seoul National University Biomedical Research Center, Seoul 03080, Korea

**Keywords:** tonsillectomy, coblation, electrocautery, postoperative pain, randomized controlled trial

## Abstract

Many surgical instruments have been introduced and compared for safety and surgical efficiency in tonsillectomy. This study aimed to compare postoperative pain between coblation and conventional monopolar electrocautery tonsillectomy by intrapatient comparison in children. Thirty pediatric patients over six years of age undergoing tonsillectomies were enrolled. Coblation and electrocautery were used to remove both tonsils in one patient; one was removed by coblation and the other by electrocautery. The coblation side was randomly selected, and it was blinded to the patients. Each side’s daily pain scores were collected on the visual analogue scale (VAS) during ten postoperative days. On the day of surgery, 22 (73%) patients identified less pain on the coblation side, while others felt similar pain. The mean pain scores were significantly lower on the coblation side during the postoperative ten days (except for the 6th and 8th) than on the electrocautery side. The duration of severe pain (VAS > 5) was significantly shorter on the coblation side than on the electrocautery side (2.0 versus 3.7 days, respectively; *p* = 0.042). Coblation tonsillectomy induced less pain than electrocautery in pediatric patients; therefore, surgeons could choose the coblator as a surgical instrument for tonsillectomy to improve the pediatric postoperative quality of life.

## 1. Introduction

Tonsillectomy is one of the common surgeries performed in otorhinolaryngology, especially for children and adolescents. Many of the surgical techniques and tools have been introduced to tonsillectomy [1]. It has been proposed that coblation tonsillectomy results in less postoperative pain and bleeding than conventional electrocautery tonsillectomy, owing to the conduction of relatively less heat in the surgical field [2,3]. Although the superiority of coblation in adenoidectomy has been confirmed in prospective multi-center studies [4,5], the advantages of coblation in tonsillectomy remain controversial. Many of the studies determined in favor of coblation in reducing postoperative pain [6,7,8,9]. However, some of the randomized studies failed to show differences in postoperative pain and bleeding [10,11,12]. A systematic review and meta-analysis of coblation versus electrocautery tonsillectomy in 2020 reported significantly reduced pain for coblation on the first postoperative day [13]; however, postoperative pain was not significantly different in a systematic review written in 2022 [14].

Because pain perception can differ among individuals, analyzing and comparing postoperative pain between two surgical methods is better accomplished in each patient. However, previously reported studies comparing coblation and electrocautery tonsillectomy performed interpatient comparisons of the two tonsillectomies from different patients with different pain perceptions; this might be related to the controversial reports on the advantage of the coblation tonsillectomy. For the first time reported in the literature, to the best of the authors’ knowledge, this study aimed to verify the differences in postoperative pain between coblation and electrocautery tonsillectomies by an intrapatient comparison of pediatric patients who can discriminate pain.

## 2. Materials and Methods

The study was conducted according to the principles expressed in the Declaration of Helsinki and was approved by the Institutional Review Board of the Biomedical Research Institute at Seoul National University Hospital (H-1312-074-541). Written informed consent was submitted by all of the subjects (or their caregivers) when they were enrolled. The randomized clinical trial was registered at the Clinical Research Information Service (CRIS, http://cris.nih.go.kr, accessed on 2 August 2022), number KCT0002922.

### 2.1. Study Design and Participants

Between February and September 2014, pediatric patients, who were scheduled to undergo tonsillectomy and adenoidectomy, were prospectively recruited to a university hospital in Korea (Seoul National University Hospital, Seoul, Korea). The pediatric patients over six years of age who could discriminate and provide answers about pain on each side of the tonsils were included in the study. The children with a history of previous adenotonsillectomy, bleeding disorders, psychological disorders, craniofacial anomaly, peri-tonsillar abscess, and neoplasm were excluded.

The study design and patient flow are illustrated in Figure 1. This study was designed as a prospective single-blinded randomized control study. The participants were blinded to the operation side of the surgical instruments. The surgeon knew the devices, but the tools were randomly selected by the randomization process described below. The information on sex and age, indication for surgery, allergic rhinitis, and sleep-related quality of life were collected. The sleep-related quality of life was evaluated by the Korean version of the Obstructive Sleep Apnea-18 questionnaire (KOSA-18, score range 18–126), where a score under 60 represents low impact, a score between 60 and 80 moderate impact, and a score over 80 significant impact on the quality of life [15].

### 2.2. Randomization and Blinding

An online statistical program (www.randomization.com, accessed on 5 February 2014) generated a random sequence to choose the surgical side for coblation and electro-cauterization. The generated random sequence was available only to a surgeon, and the surgeon followed the randomly selected instrument to operate on each tonsil. According to the random sequence, half of the patients underwent coblation tonsillectomy on the right side and electrocautery tonsillectomy on the left side; conversely, the other half underwent coblation on the left and electrocautery on the right. Both the children and the caregivers were blinded to the instruments.

### 2.3. Surgical Procedures

All of the surgeries were performed under general anesthesia in the Rose position. In the operating room, general anesthesia was induced with intravenous thiopental sodium 5 mg/kg and fentanyl 1–2 μg/kg. Endotracheal intubation was performed after a neuromuscular blockade with rocuronium 0.6 mg/kg. Anesthesia was induced with sevoflurane and maintained using desflurane. Dexamethasone 0.2 mg/kg (maximum 10 mg) and ondansetron 0.1 mg/kg (maximum 4 mg) were also administered to decrease the postoperative nausea and vomiting. The attending anesthesiologists administered additional opioids (fentanyl), as appropriate. At the end of the surgery, neuromuscular blockade was reversed with neostigmine (0.05 mg/kg) and atropine (0.02 mg/kg). No local anesthetic was applied to the surgical site.

One tonsil was removed using coblation (Evac 70 Xtra Plasma Wand, Arthrocare, Sunnyvale, CA, USA), and the other by monopolar electro-cauterization (GoldLine Electrosurgical Pencil, CONMED, Utica, NY, USA) (Figure 2). During the coblation tonsillectomy, ablation for extracapsular dissection and coagulation for bleeding control were set to levels six and three, respectively. The electrical power was set to level 13 during the electrocautery tonsillectomy for monopolar and bipolar cauterization. The following technique of extracapsular dissection was conducted for both of the tonsils under a surgical microscope. The tonsil was grasped using tonsil forceps and pulled medially. The tonsil and capsule were separated from the surrounding pharyngeal muscles and fascia (Figure 3). The dissection was conducted from the upper pole to the lower pole. The operation time for each tonsil was measured from the start of the incision to the end of bleeding control. The size of the tonsil (Brodsky grading) [16], and the levels of adhesion (mild, moderate, or severe) were also checked during surgery. One experienced otolaryngologist (D.H. Han) performed the surgeries, who had previously performed more than 1000 coblation and electrocautery tonsillectomies.

### 2.4. Postoperative Pain Assessment

After the surgery, acetaminophen (10 mg/kg) was prescribed for seven days. No other analgesic was administered in the postoperative days. All of the patients were discharged on the next day of surgery and visited the hospital ten days later. A visual analogue scale (VAS) was used to evaluate the severity of postoperative pain on both sides separately. The children and their caregivers checked the postoperative pain score in VAS (0–10) from the day of surgery through the 10th postoperative day, at 1 h after the surgery (on the operation day), and in the morning (from postoperative days 1 to 10). In this study, VAS > 5 was defined as severe postoperative pain.

### 2.5. Statistical Analysis

R version 3.5.1 (The R Foundation for Statistical Computing, Vienna, Austria) was used for statistical analyses and figures. The sample size of 30 cases for analysis was set after calculation by the daily mean pain score and standard deviation (superiority test in two populations, type 1 error (α): 5%, power = 80%). Because the operation times on the coblation side were not normally distributed, a comparison of operation times on both sides was tested, using the Wilcoxon signed-rank test. The postoperative VAS scores were normally distributed, and they were compared using the paired *t*-test. The severity of pain was compared using the χ2 test. The multivariate analyses were completed by multivariate linear regression. In all of the statistical tests, *p* < 0.05 was considered statistically significant.

## 3. Results

### 3.1. Demographic Data

A total of 32 patients were initially enrolled; however, two patients were excluded. One child failed to report daily pain scores; the other underwent additional surgery to control immediate postoperative bleeding under general anesthesia on the day of surgery (Figure 1). After excluding the two cases, the data of 30 patients were analyzed. No severe complication was noted after the surgeries among the enrolled patients. One experienced postoperative bleeding, however, the bleeding site was not found, and the bleeding was controlled with conservative management. Two-thirds of them were boys, and the others were girls (Table 1). The median age of the patients was 8.0 years (25% and 75% quartiles of age were 7.0 and 10.8 years, respectively). Most of the children experienced sleep-disordered breathing. Twenty (66.7%) children had allergic rhinitis, and the median score on the KOSA-18 was 53.5 (25% and 75% quantiles of KOSA scores were 46.0 and 61.3, respectively). Most of the participants had high grades (three–four) of tonsils, and all of the children had the same sizes of right and left tonsils (Table 2). Only one child had different levels of tonsillar adhesion to the pharyngeal muscle (severe adhesion on the coblation side and moderate adhesion on the electrocautery side).

### 3.2. Outcomes

The median duration of the operation for the coblation tonsillectomy was significantly shorter than the electrocautery tonsillectomy (median 102.5 s (25–75% quartiles: 79.3–113.4 s) and 270.5 s (168.0–344.5), respectively; *p* < 0.001). In the multivariate analysis, while the operation time on the coblation side did not correlate with sex, the operation time on the electrocautery side was longer in boys than in girls (Appendix A). Tonsillar adhesion to the pharyngeal muscle was correlated with operation time for the coblation and the electrocautery tonsillectomies: a strong correlation was found with a correlation coefficient R = 0.785 with *p* < 0.001 on the coblation side and R = 0.668 with *p* < 0.001 on the electrocautery side.

The mean VAS scores were lower on the coblation side than on the electrocautery side during the ten postoperative days (Table 3). Significant differences were noted from the day of surgery through the postoperative 10th day, except for the 6th and 8th day (Figure 4). In addition, the mean total score for the ten days of coblation tonsillectomy was significantly lower than that of electrocautery (34.9 ± 19.0 vs. 47.7 ± 23.3, *p* = 0.023). The difference in the pain scores was the highest on the day of surgery; the difference in pain scores tended to decrease over time (Figure 4). More than one-half of the children reported that the coblation side was less painful on the day of surgery and postoperative days 3 and 5 (Figure 5). On days 9 and 10, the postoperative pain was similar on both sides in most of the patients. The postoperative days with severe pain (VAS > 5) were 2.0 ± 2.5 days (mean ± standard deviation) for the coblation side and 3.7 ± 3.9 days for the electrocautery side, which was significantly different (*p* = 0.042). The daily VAS scores in the patients with sleep-disordered breathing (*n* = 26) are separately analyzed in Appendix A.

## 4. Discussion

This study was a randomized and single-blinded investigation examining the postoperative pain of coblation and electrocautery tonsillectomy in children and adolescents. In particular, this study used the intrapatient comparison of postoperative pain, which is different from conventional interpatient comparisons. Previous studies have evaluated an interpatient comparison (both of the tonsils were removed by the same instrument, coblation or electro-cauterization) without considering the individual differences in the pain threshold, which might have distorted the results and weakened the statistical differences. Single blinding resulted in the children and the caregivers not knowing which tonsil was excised by coblation or electrocautery. The intrapatient comparison and the single blinding can reduce the study bias and strengthen the results in this study.

Because adult tonsils have increased fibrosis of the tonsillar capsule and a large diameter of the blood vessels, adult tonsillectomy is often accompanied by extra-capsular muscular and neuronal injuries, with frequent intraoperative bleeding. On the other hand, in pediatric tonsillectomy, dissecting the palatine tonsils from underlying muscle is relatively easy to perform without muscular injury, and the intraoperative bleeding is minimal. Therefore, pediatric tonsils can be better candidates for evaluating tissue damage caused by surgical instruments than adult ones. Because preschool children under six years of age may not distinguish pain on each surgical side, this study included those over six years of age who were able to discriminate and express pain. Although the reduced postoperative pain of coblation compared to electrocautery tonsillectomy in pediatrics has been previously reported in a systematic review and meta-analysis, the study was not based on intrapatient comparisons, as our study was [17].

Postoperative pain after tonsillectomy is an important matter, which is related to the functional recovery of the patient. A decrease in postoperative pain is known to promote an increase in oral intake [6,18], earlier return to school [19], and reduction in analgesic medications after tonsillectomy [20]. Significantly less pain associated with coblation than electrocautery was revealed during the five consecutive postoperative days. Early postoperative pain scores were lower on the coblation side than on the electrocautery side, and most of the children reported that the coblation side was less painful. Less painful days were reported on the coblation side than on the electrocautery side. Coblation, which uses a bipolar electrical radiofrequency current through a medium, such as normal saline, operates at a relatively low tissue temperature (between 60 °C and 70 °C) compared with electrocautery (between 400 °C and 600 °C). The low temperature reduces the collateral damage and results in less postoperative pain [3,6]. The stream of saline during the coblation tonsillectomy may also cool down the surrounding tissues. The low tissue temperatures associated with coblation have yielded earlier healing of the tonsillar fossa in a rat model and human wounds [21,22].

Along with the lower tissue temperatures induced by the coblation tonsillectomy, our hypothesis is that less postoperative pain could also be attributed to the shorter operation time in coblation, resulting in a shorter duration of heat exposure to nearby tissues. The operating time of the coblation tonsillectomy was significantly faster than the electrocautery in our study. The significant factors influencing the operation time in the multivariate analysis were three parameters, which are the male sex and surgical indication in the electrocautery group, and the tonsillar adhesion grade in both of the groups. Unlike the electrocautery tonsillectomy, the operation time of the coblation tonsillectomy was not significantly increased in the male or the recurrent tonsillitis group. Nevertheless, the degree of tonsillar adhesion increased the surgical time in both of the groups. However, some previous studies have reported longer operation times in coblation tonsillectomies [22,23].

Postoperative hemorrhage in our study occurred in 2 of 32 (6.3%) cases (both boys), which is similar to previously reported rates, which ranged from 2.1% to 22.2% [24]. A 7-year-old boy experienced tonsillar bleeding on the coblation side several hours after the surgery and underwent bleeding control under general anesthesia on the day of surgery. The other child experienced oral bleeding approximately one week after surgery, controlled by conservative management. The bleeding site was not identified, and he did not require transfusion.

This study was designed to objectively compare the postoperative pain between two surgical instruments; nevertheless, there were limitations. Because the surgical skill for the coblation tonsillectomy can be different among surgeons, the operation time and the extracapsular damage that lead to postoperative pain can be increased in other situations. Another limitation of this study is that the correlation between pain and surgical technique is not evaluated for each surgical indication group, i.e., the recurrent tonsillitis group and the sleep-disordered breathing group, while the difference in adhesion or fibrosis of the peritonsillar tissue can also affect the postoperative pain. In addition, the visual analogue scale presented by the children may not precisely reflect the real level of pain. Lastly, the patient population in this study is relatively small. This study did not have sufficient participants to validate the differences between postoperative complications, especially hemorrhage. Because the incidence of postoperative bleeding is rare, hundreds of participants would have been necessary. Nevertheless, we found that postoperative hemorrhage did not exceed the previously reported rates in the literature [24].

To conclude, this randomized controlled trial and the intrapatient comparison revealed that the coblation tonsillectomy could reduce children’s postoperative pain, as compared to the monopolar electrocautery tonsillectomy. Although the pain eventually subsided to similar levels on both the coblation and electrocautery sides, the children experienced less pain during the early postoperative period (from the day of surgery through the 5th postoperative day). Surgeons might consider coblation a safe and less painful surgical technique for pediatric tonsillectomy.

## Figures and Tables

**Figure 1 jcm-11-04561-f001:**
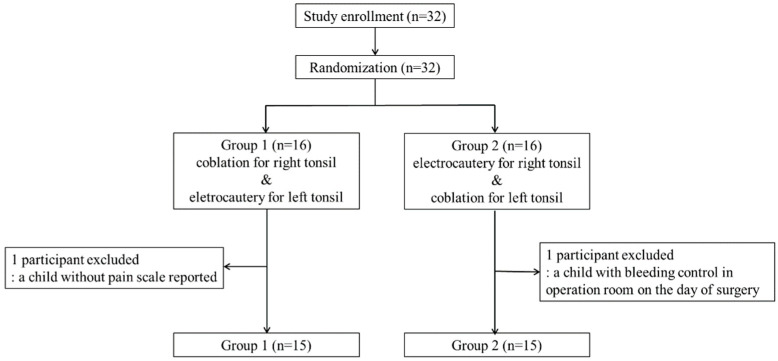
CONSORT flow diagram. The side for coblation tonsillectomy is randomly chosen to result in half of the participants undergoing coblation tonsillectomy on the left side tonsil and the other half on the right.

**Figure 2 jcm-11-04561-f002:**
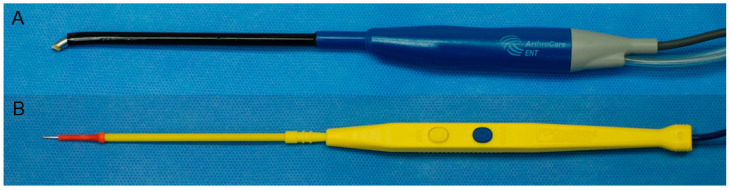
Surgical instruments used for coblation and electrocautery tonsillectomy, respectively. (**A**) Coblation hand-piece; (**B**) monopolar electrocautery hand-piece.

**Figure 3 jcm-11-04561-f003:**
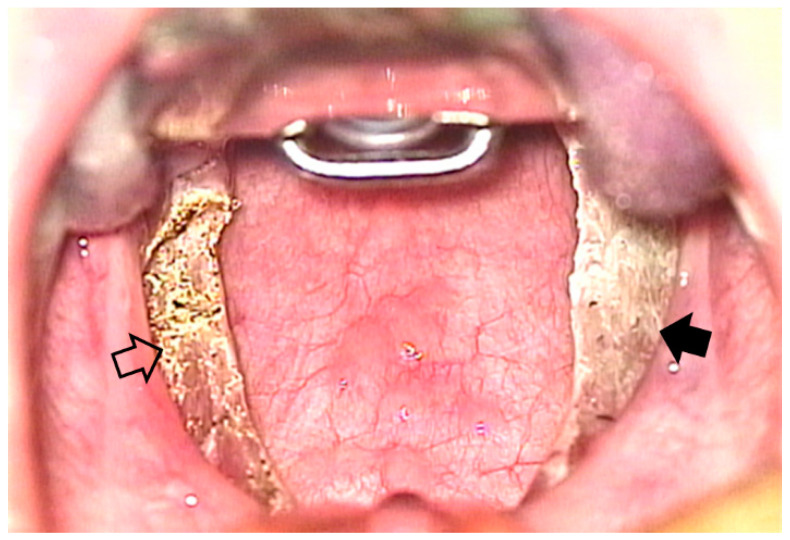
Surgical view after coblation and electrocautery tonsillectomy. Coblation and electrocautery tonsillectomy are performed on the right (solid arrow) and left (open arrow) tonsils, respectively.

**Figure 4 jcm-11-04561-f004:**
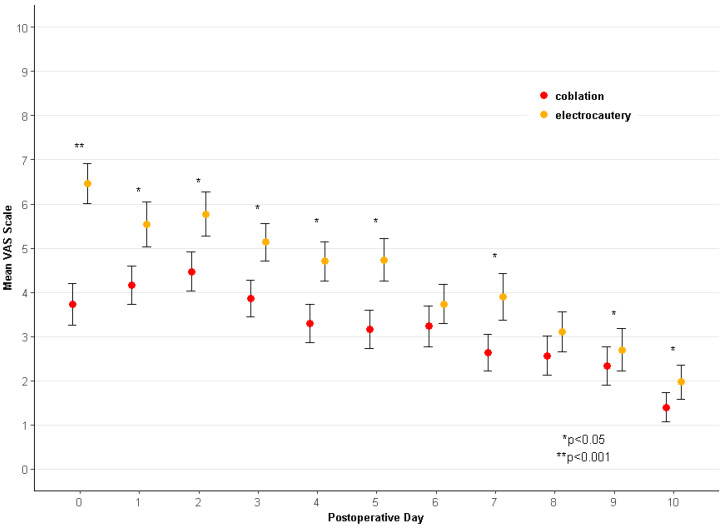
Mean postoperative pain scores according to visual analogue scale (VAS). Significantly reduced VAS pain scores are noted on the coblation side compared to the electrocautery side during postoperative 10 days, except days 6 and 8.

**Figure 5 jcm-11-04561-f005:**
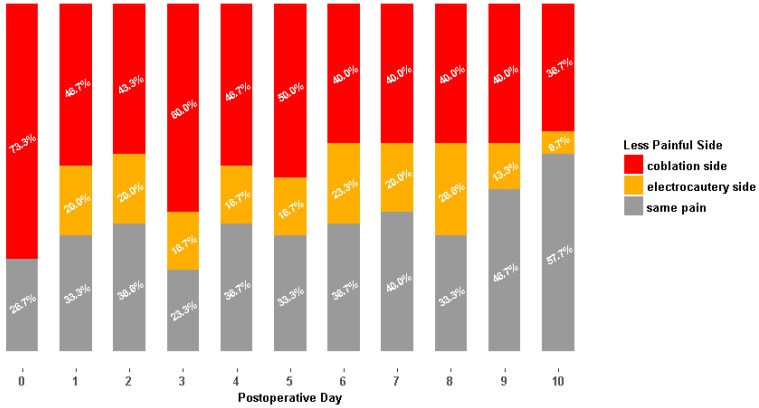
Less painful side (visual analogue scale ≤ 5). Majority of patients reporting less pain on the coblation side from the day of surgery through postoperative day 8 and similar pain on both sides on postoperative days 9 and 10.

**Table 1 jcm-11-04561-t001:** Clinical characteristics of the enrolled participants.

Characteristic	Number of Participants, *n* (%)
Total	30
Sex	
Female	10 (33.3)
Male	20 (66.7)
Age, years	
6−7	12 (40.0)
8−10	10 (33.3)
11−17	8 (26.7)
Indication of surgery	
Sleep-disordered breathing	26 (86.7)
Recurrent tonsillitis	4 (13.3)
Allergic rhinitis	
No	10 (33.3)
Yes	20 (66.7)
KOSA-18 score	
<60	22 (77.3)
60–80	6 (20.0)
≥80	2 (6.7)

Abbreviations: KOSA-18—Korean version of the Obstructive Sleep Apnea-18 questionnaire.

**Table 2 jcm-11-04561-t002:** Size and adhesion level of tonsils.

Characteristic	Coblation Side, *n* (%)	Electrocautery Side, *n* (%)
Tonsil size		
Grade 1	0 (0)	0 (0)
Grade 2	2 (6.7)	2 (6.7)
Grade 3	11 (36.7)	11 (36.7)
Grade 4	17 (56.7)	17 (56.7)
Tonsillar adhesion		
Mild	9 (30.0)	9 (30.0)
Moderate	15 (50.0)	16 (53.3)
Severe	6 (20.0)	5 (16.7)

**Table 3 jcm-11-04561-t003:** Daily pain scores, visual analog scale, on the coblation and electrocautery tonsillectomy sides (*n* = 30).

Postoperative Day	Coblation Side, VAS	Electrocautery Side, VAS	*p*-Value
0	3.7 ± 0.5	6.5 ± 0.5	<0.001
1	4.2 ± 0.4	5.5 ± 0.5	0.022
2	4.5 ± 0.4	5.8 ± 0.5	0.010
3	3.9 ± 0.4	5.1 ± 0.4	0.003
4	3.3 ± 0.4	4.7 ± 0.5	0.004
5	3.2 ± 0.4	4.7 ± 0.5	0.003
6	3.2 ± 0.5	3.7 ± 0.4	0.332
7	2.6 ± 0.4	3.9 ± 0.5	0.030
8	2.6 ± 0.4	3.1 ± 0.5	0.161
9	2.3 ± 0.4	2.7 ± 0.5	0.025
10	1.4 ± 0.3	2.0 ± 0.4	0.032

The values are presented as mean ± standard error. Abbreviations: VAS—Visual Analogue Scale.

## Data Availability

The data presented in this study are available on request from the corresponding author.

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
