# Peer review of "Intrapatient Comparison of Coblation versus Electrocautery Tonsillectomy in Children: A Randomized, Controlled Trial"

_jcm, 2022, doi:10.3390/jcm11154561_

Round 1

Reviewer 1 Report

I would like to congratulate the authors on providing a very well written report of their study. I have no negative criticisms of the grammar/syntax or structure.

The overall study design is not novel and the authors were able to achieve statistical significance for answering their hypothesis. 

Introduction: well written and offers the appropriate background for why the study question is important. 

Methods: well written, but I would recommend addressing all perioperaitve anesthetics and analgesics received, or potentially received. What is the normal perioperative pathway? Do patients receive gaseous or intravenous anesthetics? Is local anesthetic applied to the tonsillar bed and if so, when? Do the patients receive any systemic steroids or antiemetics? Do the patients receive any narcotics? Is acetaminophen the only analgesic used postoperatively? Is there potential for others?

Results: well written.  A subgroup analysis by indication would be help to find if the correlation between pain and technique differs.  While this may not be possible for recurrent tonsillitis group, it should be possible for the SDB group. It is unclear why the multivariate analysis is important for the study and there is limited discussion later on this. 

Discussion: overall very well written and addresses many important aspects of pediatric tonsillectomy. I would recommend further discussion on the importance of timing, and why such detailed statistical analysis was performed. Finally, the limitations of the study are not addressed sufficiently. Suggesting the expert level of the operating surgeon is a limitation is not appropriate. Please describe all limiting factors. 

Conclusion: an appropriate conclusive statement was made, however, I would suggest rewording it to: "..., COMPARED TO MONOPOLAR ELECTROCAUTERY, coblation tonsillectomy could reduce children's postoperative pain after tonsillectomy, as compared 

Author Response

I would like to congratulate the authors on providing a very well written report of their study. I have no negative criticisms of the grammar/syntax or structure.

The overall study design is not novel and the authors were able to achieve statistical significance for answering their hypothesis. 

Response: Thank you for your constructive comments. These comments helped us improve the quality of our manuscript. We have revised our manuscript in accordance with your recommendations and concerns. All revisions to the manuscript can be checked using the “Track Changes” function of MS Word.

Introduction: well written and offers the appropriate background for why the study question is important. 

Methods: well written, but I would recommend addressing all perioperaitve anesthetics and analgesics received, or potentially received. What is the normal perioperative pathway? Do patients receive gaseous or intravenous anesthetics? Is local anesthetic applied to the tonsillar bed and if so, when? Do the patients receive any systemic steroids or antiemetics? Do the patients receive any narcotics? Is acetaminophen the only analgesic used postoperatively? Is there potential for others?

Response: As per your recommendation, we have added all perioperative anesthetics and analgesics in the manuscript. Local anesthesia was not applied to the tonsillar bed during the surgery, and only acetaminophen was used in the postoperative days.

Revised page No. 3 Line 98

In the operating room, general anesthesia was induced with intravenous thiopental sodium 5 mg/kg and fentanyl 1–2 μg/kg. Endotracheal intubation was performed after neuromuscular blockade with rocuronium 0.6 mg/kg. Anesthesia was induced with sevoflurane and maintained using desflurane. Dexamethasone 0.2 mg/kg (maximum 10 mg) and ondansetron 0.1 mg/kg (maximum 4 mg) were also administered to decrease postoperative nausea and vomiting. The attending anesthesiologists administered additional opioids (fentanyl), as appropriate. At the end of the surgery, neuromuscular blockade was reversed with neostigmine (0.05 mg/kg) and atropine (0.02 mg/kg). No local anesthetic was applied to the surgical site.

Revised page No. 4 Line 133

No other analgesic was administered in the postoperative days.

Results: well written.  A subgroup analysis by indication would be help to find if the correlation between pain and technique differs.  While this may not be possible for recurrent tonsillitis group, it should be possible for the SDB group. It is unclear why the multivariate analysis is important for the study and there is limited discussion later on this. 

Response: Thank you for your advice. A subgroup analysis has been added for the SDB group as a Table in the supplementary file (Supplementary Table 2). A shorter operation time might have affected the less postoperative pain in the coblation group. However, as your comment, the operation time is not directly related to the topic of our study and the multivariate analysis is not important. Thus, Table 3 has been transferred as a supplementary Table 1 in the supplementary file. In addition, we have added a little bit of discussion for the multivariate analysis as below.

Revised page No. 6 Line 194

The daily VAS scores in the patients with sleep-disordered breathing (n = 26) are separately analyzed in Supplementary Table 2.

Supplementary Table 2. Daily pain scores, visual analog scale, on the coblation and electrocautery tonsillectomy sides in patients with sleep-disordered breathing (n = 26).

Postoperative day

Coblation side, VAS

Electrocautery side, VAS

p-value

0

4.0 ± 0.5

6.4 ± 0.5

< 0.001

1

4.1 ± 0.5

5.3 ± 0.6

0.064

2

4.5 ± 0.5

5.8 ± 0.6

0.022

3

3.9 ± 0.5

5.2 ± 0.5

0.006

4

3.2 ± 0.5

4.6 ± 0.5

0.007

5

3.3 ± 0.5

4.8 ± 0.5

0.004

6

3.4 ± 0.5

3.6 ± 0.4

0.647

7

2.8 ± 0.5

3.9 ± 0.5

0.060

8

2.4 ± 0.4

3.1 ± 0.5

0.065

9

2.2 ± 0.4

2.6 ± 0.5

0.036

10

1.5 ± 0.4

2.0 ± 0.4

0.091

Revised page No. 8 Line 249

Significant factors influencing the operation time in the multivariate analysis were three parameters which are the male sex and surgical indication in the electrocautery group, and the tonsillar adhesion grade in both groups. Unlike the electrocautery tonsillectomy, operation time of coblation tonsillectomy was not significantly increased in the male or recurrent tonsillitis group. Nevertheless, the degree of tonsillar adhesion increased the surgical time in both groups.

Discussion: overall very well written and addresses many important aspects of pediatric tonsillectomy. I would recommend further discussion on the importance of timing, and why such detailed statistical analysis was performed. Finally, the limitations of the study are not addressed sufficiently. Suggesting the expert level of the operating surgeon is a limitation is not appropriate. Please describe all limiting factors. 

Response: As we mentioned above, further discussion on the operation time has been added. For the limitation of the study, the expert level of the surgeon has been deleted and all the other limiting factors we know have been added (difference in operation time and the extracapsular damage, lack of subgroup analysis, the VAS score, and the small sample size).

Revised page No. 8 Line 246

Along with the lower tissue temperatures induced by the coblation tonsillectomy, our hypothesis is that less postoperative pain could also be attributed to the short operation time in coblation, resulting in a shorter duration of heat exposure to nearby tissue.

Revised page No. 8 Line 265

Because the surgical skill for the coblation tonsillectomy can be different among surgeons, the operation time and the extracapsular damage that lead to postoperative pain can be increased in other situations. Another limitation of this study is that the correlation between pain and surgical technique is not evaluated for each surgical indication group, i.e. the recurrent tonsillitis group and the sleep-disordered breathing group, while the difference in adhesion or fibrosis of the peritonsillar tissue can also affect the postoperative pain. In addition, the visual analogue scale presented by the children may not reflect the real level of pain precisely. Lastly, the patient population in this study is relatively small.

Conclusion: an appropriate conclusive statement was made, however, I would suggest rewording it to: "..., COMPARED TO MONOPOLAR ELECTROCAUTERY, coblation tonsillectomy could reduce children's postoperative pain after tonsillectomy, as compared 

Response: Thank you for pointing out our omission. We have added "COMPARED TO MONOPOLAR ELECTROCAUTERY” in the sentence below.

Revised page No. 9 Line 278

To conclude, this randomized controlled trial and the intrapatient comparison revealed that coblation tonsillectomy could reduce children's postoperative pain, as compared to monopolar electrocautery tonsillectomy.

Reviewer 2 Report

The authors present a randomized blinded control trial  examining outcomes of   intracapsular collation tonsillectomy vs. traditional electrocautery tonsillectomy in terms of postoperative pain and bleeding. This study is novel in that it uses patients as their own controls. The findings are limited by small sample size. However, the study was well designed and executed and the results show significant improvement in postoperative pain in the early days after tonsillectomy and no significant increase in postoperative bleeding in this sample. The authors acknowledge that this study is not powered to detect differences in postoperative complications. As patients were their own controls, it likely does not matter that the population was not homogenous in their indication for tonsillectomy (sleep disordered breathing and tonsillitis)

Question for authors:

1. did the decrease in postoperative pain for the first 5 days of surgery translate to increased function (ie. increased oral intake or early return to school or less pain medication administration)?

2. I would remove the entire paragraph comparing pediatric and adult tonsillectomy as well as the (lines 50-61). There are many randomized controlled trials demonstrating an improvement in pain in adults undergoing intracapsular tonsillectomy with coblation wand.  

Author Response

  1. did the decrease in postoperative pain for the first 5 days of surgery translate to increased function (ie. increased oral intake or early return to school or less pain medication administration)?

Response: First of all, thank you for your precious comments on our manuscript. Especially, this point for the increased function by the decreased pain is an important issue that we missed in the previous manuscript. Although the amount of oral intake, return to school, and the amount of pain medication was not compared in this study, we found relevant reports in the literature that decreased postoperative pain ensures increased function after surgery and have added this issue in the Discussion section of the revised manuscript. Once again, thank you for the valuable comment.

Revised page No. 8 Line 231

Postoperative pain after tonsillectomy is an important matter, which is related to the functional recovery of the patient. Decreased postoperative pain is known to promote an increase in oral intake [6,18], earlier return to school [19], and reduction in analgesic medications after tonsillectomy [20].

  1. I would remove the entire paragraph comparing pediatric and adult tonsillectomy as well as the (lines 50-61). There are many randomized controlled trials demonstrating an improvement in pain in adults undergoing intracapsular tonsillectomy with coblation wand.  

Response: We have deleted the entire paragraph as your recommendation. Thank you for your comments.